# One-Pot Biocatalytic Preparation of Enantiopure Unusual α-Amino Acids from α-Hydroxy Acids via a Hydrogen-Borrowing Dual-Enzyme Cascade

**Fei Liu †, Junping Zhou †, Meijuan Xu, Taowei Yang, Minglong Shao, Xian Zhang \***
**and Zhiming Rao \***

The Key Laboratory of Industrial Biotechnology, Ministry of Education, School of Biotechnology, Jiangnan University, Wuxi 214122, China; liufei1110@outlook.com (F.L.); zhoujp116@sina.com (J.Z.); xumeijuan@jiangnan.edu.cn (M.X.); yangtw@jiangnan.edu.cn (T.Y.); mlshao@jiangnan.edu.cn (M.S.)

\* Correspondence: zx@jiangnan.edu.cn (X.Z.); raozhm@jiangnan.edu.cn (Z.R.); Tel.: +86-0510-85916881 (Z.R.); Fax: +86-0510-85918516 (Z.R.)

† Authors contributed equally to this study.

**Abstract:** Unusual α-amino acids (UAAs) are important fundamental building blocks and play a key role in medicinal chemistry. Here, we constructed a hydrogen-borrowing dual-enzyme cascade for efficient synthesis of UAAs from α-hydroxy acids (α-HAs). D-mandelate dehydrogenase from *Lactobacillus brevis* (*Lb*MDH) was screened for the catalysis of α-HAs to α-keto acids but with low activity towards aliphatic α-HAs. Therefore, we rational engineered *Lb*MDH to improve its activity towards aliphatic α-HAs. The substitution of residue Leu243 located in the substrate entrance channel with nonpolar amino acids like Met, Trp, and Ile significantly influenced the enzyme activity towards different α-HAs. Compared with wild type (WT), variant L243W showed 103 U/mg activity towards D-α-hydroxybutyric acid, 1.7 times of the WT's 60.2 U/mg, while its activity towards D-mandelic acid decreased. Variant L243M showed 2.3 times activity towards D-mandelic acid compared to WT, and its half-life at 40 °C increased to 150.2 h comparing with 98.5 h of WT. By combining *Lb*MDH with L-leucine dehydrogenase from *Bacillus cereus*, the synthesis of structurally diverse range of UAAs from α-HAs was constructed. We achieved 90.7% conversion for L-phenylglycine production and 66.7% conversion for L-α-aminobutyric acid production. This redox self-sufficient cascade provided high catalytic efficiency and generated pure products.

**Keywords:** D-mandelate dehydrogenase; unusual α-amino acids; rational engineering; site-saturation mutagenesis; hydrogen-borrowing cascade

## 1. Introduction

Unusual amino acids (UAAs) with readily functionalized amine and carboxyl groups are valuable building blocks of modern medicinal chemistry [1–3], including free amino acid drugs, anticancer agents, antimicrobial peptides, and cyclic peptide antibiotics, among others [4–7]. A large amount of drugs and biologically active molecules rely on central UAAs [8,9], such as L-phenylglycine, which has been applied in β-lactam antibiotics including penicillin and pristinamycin I [10,11], while L-α-aminobutyric acid is an important precursor for the synthesis of many chiral drugs like brivaracetam and ethambutol [12,13]. Furthermore, UAAs also have great potential in the preparation of smart nanomaterials, for example, part of dipeptides containing norvaline can be assembled into microporous organic materials [1,14]. Currently, the preparation of UAAs is mainly achieved by chemical synthesis which are environmentally unfriendly and result in many byproducts [15,16]. Hence, it is necessary to produce UAAs by biocatalytic processes.

One-pot biocatalytic preparation of high-value chemicals is an essential strategy, with the advantage of concise reaction, excellent enantioselectivity, and environmental friendliness [17]. In recent years, two or multi-enzyme cascade catalysis systems have been widely used in the synthesis of UAAs [18]. Li et al assembled eight enzymes for the production of aromatic $\alpha$-amino acids from original aromatic alkene substrates [19]. While the substrates are constraint to aromatic alkene, substitution of –X, -Me, and –OMe on benzene ring results in phenylglycine or its derivative as the products; however, this cascade system required cosubstrate NAD(P)H and substrate $O_2$ which restricted its catalytic efficiency. It was reported that the introduction of a three-enzyme redox-neutral cascade improved production of L-phenylglycine from racemic mandelic acid, whereby the hydride liberated in the oxidation of mandelic acid to benzoylformic acid was directly consumed in the reductive amination of benzoylformic acid using NAD as a cofactor [20]. This cascade is also known as a hydrogen-borrowing cascade [21]. In addition, studies reported the use of the same cascade with some novel D-mandelate dehydrogenases for the production of L-phenylglycine with high catalytic efficiency and good enantioselectivity [22,23]. Therefore, it is an efficient strategy for synthesis of UAAs using D-$\alpha$-hydroxy acid (D-$\alpha$-HA) substrates through hydrogen-borrowing cascade reactions. However, in these studies, most of the multi-enzyme cascade systems are used for the synthesis of aromatic products, and it is still a challenge for the biocatalytic preparation of other groups of UAAs like aliphatic UAAs.

Screening novel D-hydroxy acid dehydrogenases with high activity towards aliphatic D-$\alpha$-HAs is the ideal approach, however, the natural enzymes usually do not achieve high productivity, making engineering of known enzymes of prime importance. Directed evolution, rational, and semi-rational engineering are common strategies for enzymatic modification [24–27]. Compared to rational engineering, high-through screening methods need plenty of work but usually have very few positive mutations. By rational engineering and analyzing the substrate channel, Bao et al. identified residues which control the substrate chain-length selectivity of cyanobacterial aldehyde-deformylating oxygenase [28]. After structural and substrate channel analysis, Song et al. used semi-saturated mutation for expanding the substrate tunnel and obtained two mutations with 18-fold increase in the initial reaction rate [16]. Therefore, it is an efficient strategy for changing substrate specificity by expanding or narrowing the substrate channel.

Here, we screened a NAD-dependent D-mandelate dehydrogenase from *Lactobacillus brevis* (*Lb*MDH) which was identified by Fan et al. and found that its activity towards aliphatic substrates like D-$\alpha$-hydroxybutyric acid was low. Thus, rational engineering of *Lb*MDH was conducted by engineering the substrate entrance tunnel to improve the substrate affinities and catalytic efficiencies. Then, we introduced NADH-dependent L-leucine dehydrogenase from *Bacillus cereus* (*Bc*LeuDH) to construct a hydrogen-borrowing dual-enzyme cascade system for catalyzing D-$\alpha$-HAs to L-UAAs (shown in Figure 1). As in our precious work, we found that *Bc*LeuDH could catalyze not only aromatic substrates but also aliphatic $\alpha$-keto acids [9,29], which was suitable for this cascade system. Our strategy provided an efficient strategy for biocatalytic preparation of a broad-range L-UAAs, with the enhancement of *Lb*MDH activity towards aromatic and aliphatic $\alpha$-Has, contributing significantly to the biocatalytic preparation of UAAs.

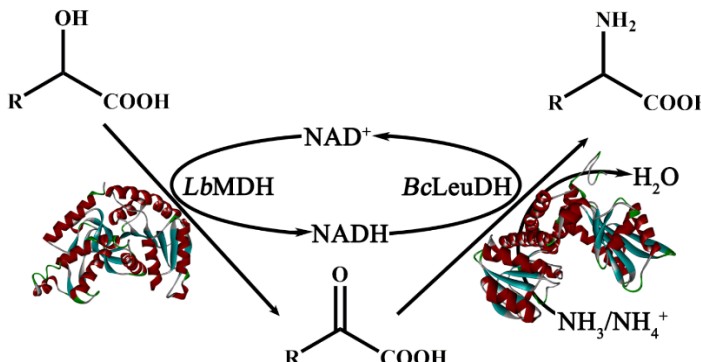

**Figure 1.** Scheme of the asymmetric biocatalytic preparation for UAAs from α-HAs via a hydrogen-borrowing dual-enzyme cascade composed of *L. brevis* D-mandelate dehydrogenase (*Lb*MDH) and *Bacillus cereus* L-leucine dehydrogenase (*Bc*LeuDH).

## 2. Results and Discussion

### 2.1. Expression and Comparative Enzyme Activity of D-Hydroxy Acid Dehydrogenases

In order to find an efficient dehydrogenase that could synthesize different α-keto acids, we screened for D-hydroxy acid dehydrogenase with high activity and broad substrate spectrum. We chose five D-hydroxy acid dehydrogenases from *L. brevis*, *Enterococcus faecalis*, *Staphylococcus aureus*, and *Pseudomonas aeruginosa* including *Lb*MDH, *Ef*2D2R, *Sa*DlacDH, *Pa*DlacDH, and *Pa*2D2R using the NCBI database and expressed them in *E. coli* BL21 (DE3). The expression of these enzymes was analyzed by SDS-PAGE (Figure S1). The purified enzyme activities and the optimum pH towards substrate D-mandelic acid, D-α-hydroxybutyrate and D-α-hydroxycaproic acid were shown in Table 1. Interestingly, they were all alkalophilic dehydrogenases (Figure S2), and, among which, D-mandelate dehydrogenase from *L. brevis* exhibited higher activity with all the substrates. The *Lb*MDH displayed the highest activity towards D-mandelic acid, with the specific enzyme activity of 243.54 U/mg using NAD as the cofactor. However, the activity for aliphatic substrates D-α-hydroxybutyrate and D-α-hydroxycaproic acid were only 24.7% and 48.8% of the activity for D-mandelic acid, respectively. Therefore, enzyme modification of *Lb*MDH was necessary in order to improve the activity towards aliphatic α-HAs.

**Table 1.** Comparison of enzyme activities towards D-α-HAs of D-hydroxy acid dehydrogenase from different sources.

| Source | Enzyme | D-Mandelic Acid (U/mg) | D-α-Hydroxybutyric Acid (U/mg) | D-α-Hydroxycaproic Acid (U/mg) | Optimum pH | Cofactor |
|---|---|---|---|---|---|---|
| *L. brevis* | *Lb*MDH | 243.5 ± 13.5 | 60.2 ± 4.7 | 118.9 ± 10.6 | 10.5 | NAD$^+$ |
| *E. faecalis* | *Ef*2D2R | 12.4 ± 1.3 | 13.2 ± 3.9 | 10.5 ± 2.4 | 10.5 | NAD$^+$ |
| *S. aureus* | *Sa*DlacDH | 6.5 ± 0.9 | n/a | n/a | 11.0 | NAD$^+$ |
| *P. aeruginosa* | *Pa*DlacDH | 0.9 ± 0.4 | n/a | n/a | 11.5 | NAD$^+$ |
| *P. aeruginosa* | *Pa*2D2R | n/a | 0.7 ± 0.2 | 0.8 ± 0.4 | 11 | NADP$^+$ |

Note: n/a represented not available in this case.

### 2.2. Rational Engineering of LbMDH

Our results indicated that the catalytic ability of *Lb*MDH significantly decreased as the size of the substrate side chain decreased. Therefore, rational engineering of *Lb*MDH towards aliphatic substrates with short side chains was necessary for further application. Firstly, the *Lb*MDH homology model was constructed by SWISS-MODEL using D-mandelate dehydrogenase from *E. faecium* as template. Sequence alignment of *Lb*MDH with other dehydrogenase from *Lactobacillus harbinensi* and *E. faecium* which had the same catalytic function (Figure S2) was then performed. To determine the residues for mutation, structural analysis between *Lb*MDH and *Ef*MDH was carried out (Figure 2A). Residue K186,

corresponding to K187 of *Ef*MDH, has been considered to be conserved and as the crucial acid/base catalyst [30]. The NADH and substrate binding sites involving R268, N104, E271, N194, S259, and N190 were also conserved—among which N104 promoted the shear motion of the N-terminal domain that promotes orientation of NADH closer to the substrate by 1.7 Å [31]. M128 and T130 in *Ef*MDH comprised the entrance of the substrate binding pocket, while the corresponding sites in *Lb*MDH were V127 and S129, respectively. Both two residues had smaller side chains compared with those in *Ef*MDH, which meant a larger pocket. With the help of structure and sequence analysis, we had a thorough understanding of the conserved active site and substrate channel of *Lb*MDH. In addition, it has been proved that the residues near the active pocket or substrate channel show a significant effect in enzymatic catalytic efficiency and the substrate affinity [9,32]. Thus, six residues (V127, L189, L193, I204, L243, and Q247) near the active pocket or substrate channel were chosen for site-directed mutagenesis (Figure 2B). We aimed at improving the catalytic efficiency towards aliphatic substrates by changing the steric hindrance; thus, Ala and Trp, which have shorter or bigger side chains, were both chosen for target amino acids of mutagenesis. As a result, the variants V127A, L189A, L189W, L193A, L193W, I204A, L243W, L243A, Q247A, and Q247W were constructed.

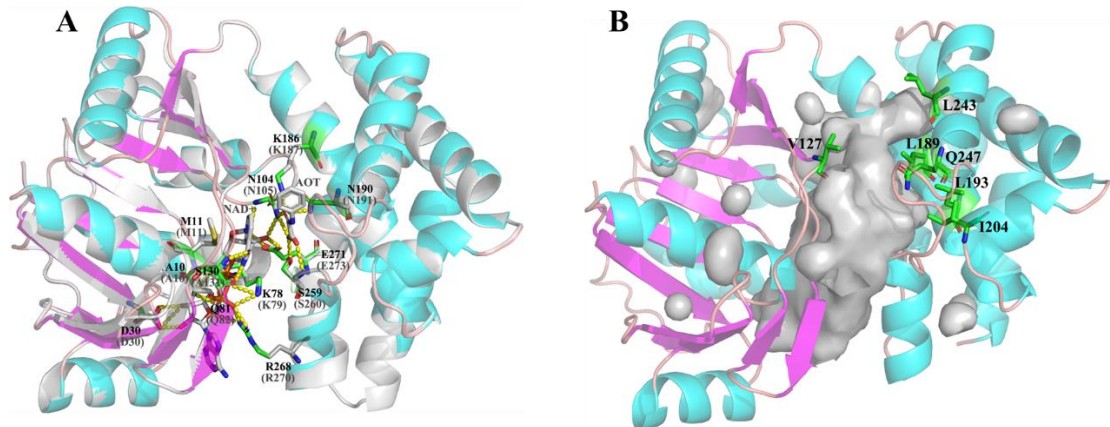

**Figure 2.** Structure analysis of *Lb*MDH. (**A**) structure analysis of alignment between *Lb*MDH and *Ef*MDH (PDB ID 5X20, showed in gray). The NAD and substrate binding site of *Ef*MDH marked in gray and the corresponding residues of *Lb*MDH marked in green were all showed as sticks with the hydrogen bonds in yellow. (**B**) Cavities and pockets of *Lb*MDH were showed in gray, and the residues near the active pocket or substrate channel were marked in green.

After purification, the relative activities were determined using D-α-hydroxybutyric acid and D-mandelic acid as substrates. As shown in Figure 3A, only variant L243W showed improved catalytic activity towards D-α-hydroxybutyric acid (approximately 1.71 times compared to that of the wild type). As D-mandelic acid is the natural substrate, the enzyme activity of L243W mutation towards D-mandelic acid was measured; unfortunately, the enzyme activity towards substrate D-mandelic acid decreased by 57.9% (Figure 3B).

### 2.3. Site-Saturation Mutagenesis at Residue L243

Since the variant L243W showed different activity towards aliphatic and aromatic substrates, L243 in *Lb*MDH was considered as a crucial residue for studying the enzyme's substrate spectrum, and the site-saturation mutagenesis was carried out at residue L243. As shown in Figure 4, the results indicated that the replacement of Leu with Phe and Trp significantly improved the activity towards D-α-hydroxybutyric acid by 52.9% and 71%, respectively. For substrate D-α-hydroxybutyric acid, variants L243H, L243I, L243M, and L243V showed relatively similar activity as the wild type. Interestingly, the substitutions that increased or had some influence on the enzyme activities towards D-α-hydroxybutyric acid were all hydrophobic amino acids, which meant that the hydrophobic

environment at this position was very important for activity. Then, the substrate spectrums of *Lb*MDH and the six mutations were studied (Table 2). It was found that the highest activity towards different substrates was observed in different variants; for example, the L243M variant showed the highest activity towards D-mandelic acid, which was 1.3 times higher than that of wild type, the L243W variant showed 71% increased activity towards D-α-hydroxybutyric acid, while L243I variant had a 46.9% increase in activity towards D-α-hydroxycaproic acid compared with wild type.

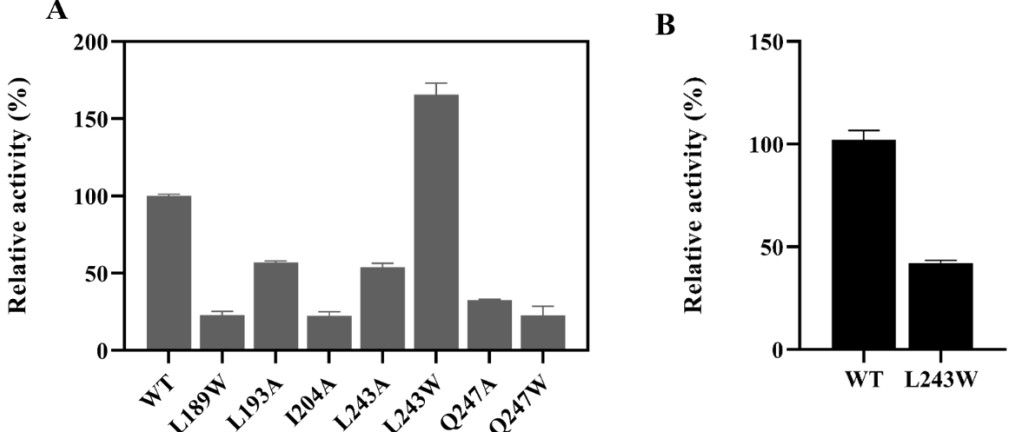

**Figure 3.** Screening of mutants. (**A**) relative activities of *Lb*MDH wild type and the variants with D-α-hydroxybutyric acid as substrate; (**B**) relative activities of *Lb*MDH wild type and L243W variant with D-mandelic acid as substrate. The enzyme activities towards D-hydroxybutanoic acid and D-mandelic acid of wild type were 243.5 and 60.2 U/mg, which were set to 100%, respectively. All assays were performed in triplicate.

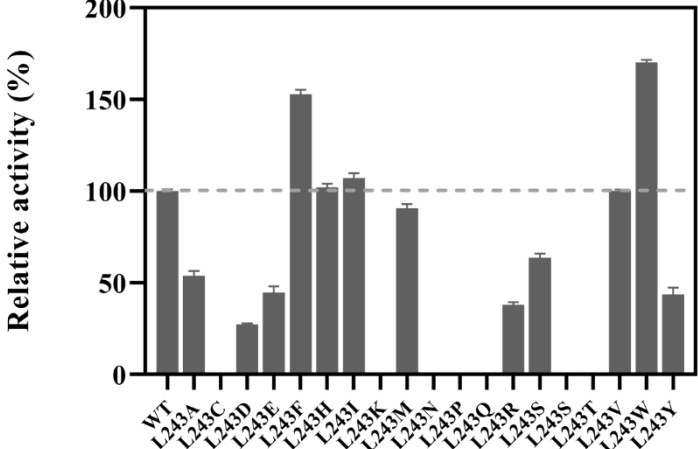

**Figure 4.** Enzyme activities of *Lb*MDH wild type and the site-saturation mutations at L243 with D-α-hydroxybutyric acid as the substrate.

## 2.4. Enzyme Characterization and Kinetic Analysis of LbMDH and Variants

To study the enzymatic characteristics, we used D-mandelic acid as the model substrate. As showed in Figure 5A, the optimum temperature for L243I was 35 °C, while the optimum temperature for the wild type and other variants were 40 °C. Above 45 °C, the enzyme activity decreased sharply and the enzyme barely had any activity at 50 °C (Figure 5A), which indicated that the enzyme was not stable at high temperatures. The half-lives of thermal inactivation ($t_{1/2}$) at 40 °C for *Lb*MDH was 98.5 h, as showed in (Figure 5B, Figure S5), while only variant L243M showed improved $t_{1/2}$ of about 140.2 h

compared to that of the wild type—whereas, the substitution of L243W and L243I in *Lb*MDH showed declined thermostabilities. Afterwards, the optimum pH of *Lb*MDH and its variants were analyzed, and they all showed the highest activity under the condition of pH 10.0–11.0 (Figure S4). For the pH stability experiments, all the enzymes were stored at 4 °C in pH 3.0–12.0 buffer for 24 h (Figure 5C). The relative activity results revealed that the enzymes were highly stable in pH range 6.0–10.0 and were unstable when the pH was under 4.0 or above 10.0, except for the L243W mutation which had lower residual activity compared to other variants.

**Table 2.** The specific activities of *Lb*MDH and its variants with different D-Hydroxy acids.

| Substrate | Relative Activity (%) | | | | | | |
|---|---|---|---|---|---|---|---|
| | WT | L243M | L243W | L243V | L243I | L243F | L243H |
| D-Mandelic acid | 100 | 231.2 ± 5.6 | 42.1 ± 0.8 | 169.5 ± 6.7 | 192.7 ± 2.5 | 50.8 ± 1.9 | 105.2 ± 2.1 |
| D-α-Hydroxybutanoic acid | 100 | 90.0 ± 1.3 | 171.0 ± 8.1 | 99.9 ± 4.2 | 107.5 ± 5.3 | 152.9 ± 9.3 | 98.3 ± 3.0 |
| D-α-hydroxycaproic acid | 100 | 121.6 ± 3.7 | 85.8 ± 6.0 | 130.8 ± 7.9 | 146.9 ± 7.8 | 93.7 ± 4.1 | 106.6 ± 3.5 |
| D-Phenyllactic acid | 100 | 78.5 ± 4.3 | 79.7 ± 3.2 | 67.0 ± 4.8 | 138.8 ± 9.2 | 81.4 ± 3.9 | 92.8 ± 6.2 |
| D-Lactic acid | 100 | 92.6 ± 8.9 | 339.1 ± 15.9 | 126.3 ± 6.3 | 75.9 ± 4.4 | 424.2 ± 29.7 | 130.5 ± 5.0 |

Note: The enzyme activity of WT for different substrates was considered to be 100%. The enzyme activity of WT for D-mandelic acid, D-α-hydroxybutanoic acid, D-α-hydroxycaproic acid, D-phenyllactic acid, and D-lactic are 243.5, 60.2, 118.9, 26.2, 1.4 U/mg.

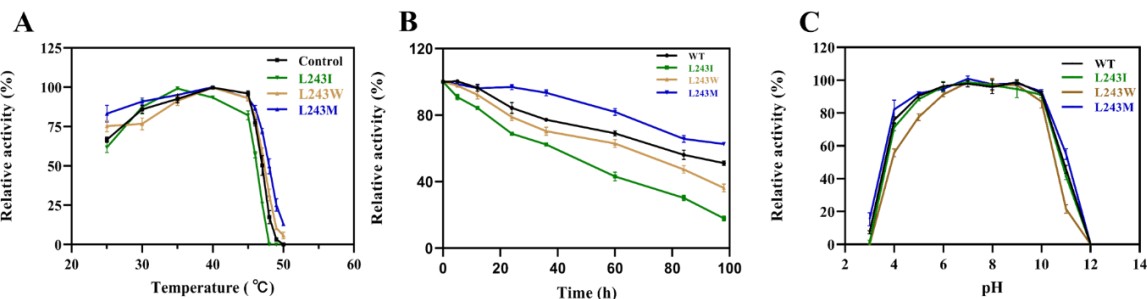

**Figure 5.** Biochemical characterization of *Lb*MDH and its variants. (**A**) effect of temperature on the enzymatic activity of *Lb*MDH and its variants; (**B**) thermostability was determined by measuring the enzymatic activity of *Lb*MDH and its variants after treatment for the corresponding time at 40 °C; (**C**) effects of pH stability on the enzymatic activity of *Lb*MDH and its variants, assayed at different pH for 24 h.

Kinetic parameters of the purified *Lb*MDH and its variants were measured using different α-HAs as the substrates (Table 3). Compared with the wild type, variant L243W showed 3.45 times lower $K_m$ with D-α-hydroxybutyric acid, and it also showed lower $K_m$ with D-lactic acid. However, variant L243W showed increased $K_m$ with D-mandelic acid, D-α-hydroxycaproic acid, and D-phenyllactic acid, while these substrates were with long side chains. Variant L243I showed lower $Km$ and improved catalytic efficiency towards D-phenyllactic acid and D-α-hydroxycaproic acid compared to the wild type. The preferred substrate of L243M was D-mandelic acid with improved $k_{cat}/K_m$ of 0.05 $S^{-1}mM^{-1}$ compared to the wild type. However, the $k_{cat}/Km$ of the variants still not high and compared with the wild type, and all $K_m$ values of the variants for cofactor NAD increased. These results indicated that mutations at L243 changed the size of substrate entrance channel, which could influence the substrate affinities and catalytic efficiencies.

## 2.5. Structure Analysis of LbMDH and Variants

Structure analysis of *Lb*MDH and the variants was carried out to study the reason for the different specific activities and affinities towards different substrates. Molecular docking of *Lb*MDH and its variants together with D-mandelic acid and cofactor NAD were carried out by Autodock vina [33]. As showed in Figure 6, residue 243 was located at the substrate entrance channel and there were no inter-molecular hydrogen bonds between residue 243 and other residues. The replacement of Leu243

with Trp narrowed the substrate entrance channel, making the enzyme substrate entrance channel more favorable for substrates with small side chains. The L243M and L243I mutations also changed the size of entrance channel making it accessible by different substrates.

**Table 3.** Kinetic parameters of *Lb*MDH and its variants.

| Substrate | WT | | | L243M | | |
|---|---|---|---|---|---|---|
| | $Km$ (mM) | $k_{cat}$ (s$^{-1}$) | $k_{cat}/Km$ (s$^{-1}$mM$^{-1}$) | $Km$ (mM) | $k_{cat}$ (s$^{-1}$) | $k_{cat}/Km$ (s$^{-1}$mM$^{-1}$) |
| NAD$^+$ | 0.25 ± 0.012 | 0.41 ± 0.004 | 1.661 | 0.45 ± 0.071 | 0.42 ± 0.027 | 0.933 |
| D-mandelic acid | 3.71 ± 0.031 | 0.44 ± 0.011 | 0.118 | 3.09 ± 0.196 | 0.52 ± 0.090 | 0.168 |
| D-α-hydroxybutanoic acid | 31.7 ± 0.580 | 0.22 ± 0.005 | 0.007 | 25.9 ± 0.207 | 0.05 ± 0.006 | 0.002 |
| D-α-hydroxycaproic acid | 20.9 ± 0.106 | 0.41 ± 0.037 | 0.02 | 18.7 ± 0.583 | 0.51 ± 0.032 | 0.027 |
| D-phenyllactic acid | 47.2 ± 0.237 | 0.16 ± 0.002 | 0.003 | 50.4 ± 1.070 | 0.13 ± 0.028 | 0.003 |
| D-lactic acid | 60.4 ± 2.934 | 0.05 ± 0.001 | 0.001 | 61.7 ± 4.535 | 0.08 ± 0.005 | 0.001 |
| Substrate | L243W | | | L243I | | |
| | $Km$ (mM) | $k_{cat}$ (s$^{-1}$) | $k_{cat}/Km$ (s$^{-1}$mM$^{-1}$) | $Km$ (mM) | $k_{cat}$ (s$^{-1}$) | $k_{cat}/Km$ (s$^{-1}$mM$^{-1}$) |
| NAD$^+$ | 0.72 ± 0.008 | 0.15 ± 0.031 | 0.038 | 0.49 ± 0.014 | 0.61 ± 0.007 | 1.245 |
| D-mandelic acid | 5.88 ± 0.119 | 0.13 ± 0.008 | 0.022 | 4.01 ± 0.193 | 0.46 ± 0.005 | 0.114 |
| D-α-hydroxybutanoic acid | 9.2 ± 0.470 | 0.11 ± 0.003 | 0.012 | 49.8 ± 1.981 | 0.24 ± 0.073 | 0.005 |
| D-α-hydroxycaproic acid | 21.5 ± 0.134 | 0.12 ± 0.022 | 0.006 | 15.8 ± 0.335 | 0.49 ± 0.006 | 0.031 |
| D-phenyllactic acid | 49.8 ± 0.576 | 0.10 ± 0.019 | 0.002 | 33.7 ± 0.985 | 0.18 ± 0.038 | 0.005 |
| D-lactic acid | 48.2 ± 1.339 | 0.13 ± 0.009 | 0.003 | 65.3 ± 1.805 | 0.09 ± 0.002 | 0.001 |

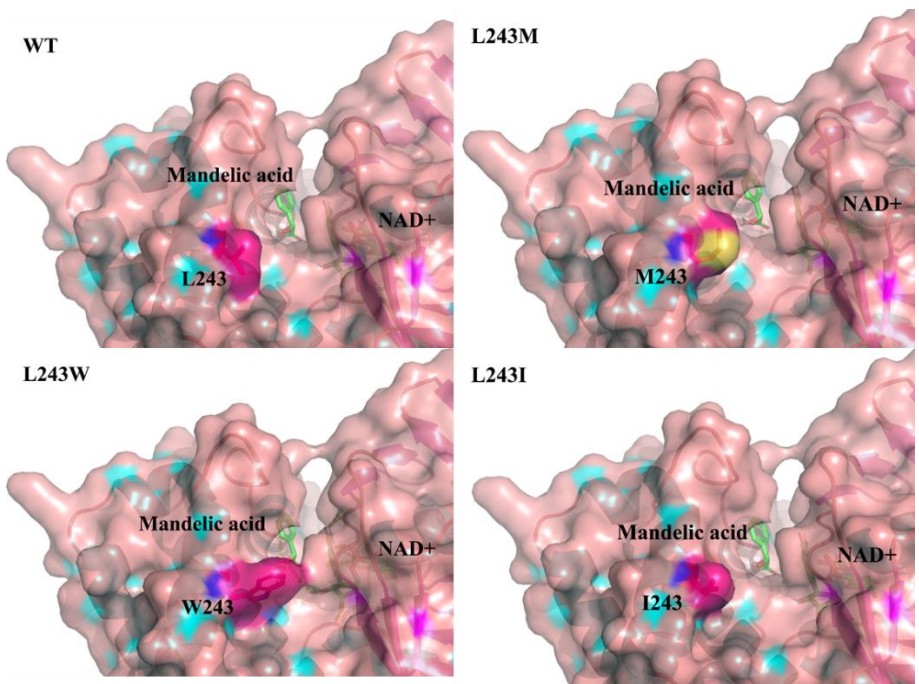

**Figure 6.** Structure analysis of *Lb*MDH and the variants. The structures of *Lb*MDH and its variants were modeled by SWISS-MODEL and displayed by PyMol. The structure was displayed with different colors: helix marked in cyan, sheet marked in magenta, loop marked in orange, ligands with the carbon backbone marked in green, and the 243th residue marked in magenta.

To further study the structural differences of *Lb*MDH and its variants, molecular dynamics was performed in 298 K for 30 ns with protein complexes including enzymes and ligands. As shown in Figure S6, the total energy value of *Lb*MDH and its variants with different substrates were almost the same, indicating that the mutations in residue 243 had little effect on the conformation stability of

the protein structure [34]. We also analyzed the root mean square fluctuation (RMSF) value (shown in Figure S6), and results showed minimal differences in the engineered structure compared to the wild type, while slight differences were observed in gray shadowed regions representing the loops far from the active center (Figure S7). Through these preliminary molecular dynamics analysis, we noticed that the mutations at 243 did not directly influence the catalytic ability, probably as the 243th residue located at the substrate channel and was distant from the active center. To further explain the effect of mutation at position 243 on the enzymatic specific activities and affinities towards different substrates, further molecular dynamics targeting the dynamic of the entrance channel could be done in later research.

### 2.6. Secondary Structure and Thermostability Analysis by Circular Dichroism (CD)

To verify the proper folding of *Lb*MDH and its variants, the secondary structure was determined by circular dichroism. As shown in Figure 7, the far-UV CD spectrum of the four enzymes exhibited similar absorbance from 190 to 240 nm, with all showing positive peaks at 193 nm and two negative peaks at 209 nm and 219 nm, suggesting correct folding of the mutants [34,35]. To further study the temperature stability of *Lb*MDH and variants, we measured the melting temperature (Tm) by monitoring their dichroic signal reduction at 220 nm at an elevated temperature from 20–70 °C. The variant L243M had a Tm value of 40.50 °C, which was 1.58 °C higher than that of the wild type, in line with the thermo-stability essay results for variant L243M.

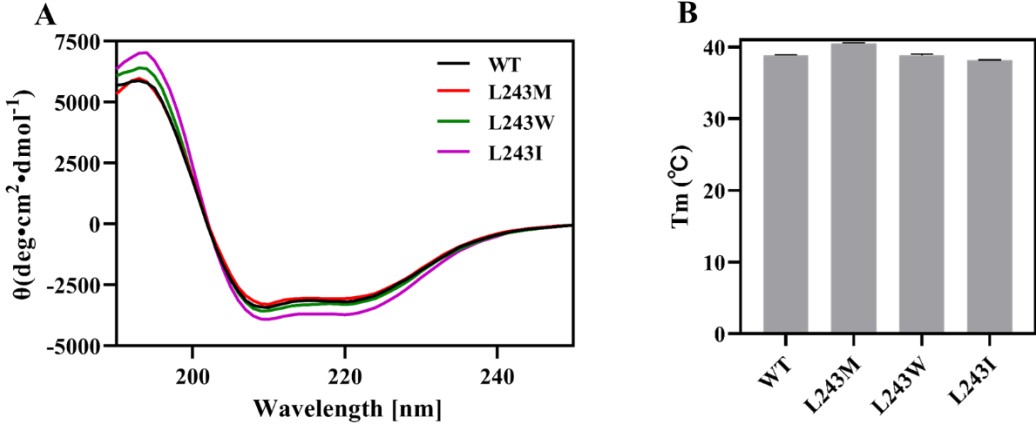

**Figure 7.** Secondary structure and melting temperature analysis (**A**) circular dichroism spectra for WT and variants; (**B**) the melting temperatures of *Lb*MDH and its variants.

### 2.7. One-Pot Biocatalytic Preparation of UAAs with a Double-Enzyme System

As UAAs are such important chemicals, the biocatalytic preparation of UAAs would be valuable, especially for the production of aliphatic and aromatic UAAs. Thus, we designed a hydrogen-borrowing dual-enzyme cascade for the synthesis of structurally diverse range of aromatic and aliphatic UAAs from $\alpha$-HAs. Here, we used the engineered *Lb*MDH together with *Bc*LeuDH$_{T45M/E116V}$ constructed before in our lab to biosynthesize UAAs [9]. Then, the dual-enzyme cascade biocatalysis preparation was conducted in a shake flask with 0.1 mg/mL *Lb*MDH or its variants and 0.5 mg/mL *Bc*LeuDH$_{T45M/E116V}$. For the biocatalytic preparation of L-phenylglycine, we used an L243M variant of *Lb*MDH, which had the highest catalytic ability towards D-mandelic acid in this study. As showed in Figure 8A, upon adding D-mandelic acid (150 mM, 22.8 g/L) to the reaction system once, the production yield of *Lb*MDH$_{L243M}$ reached 20.7 g/L, higher than the 18.9 g/L of using wild type and, most importantly, the production efficiency improved from 0.30 g/L/min to 0.47 g/L/min. Similarly, variant L243W of *Lb*MDH was used for biotransformation of D-$\alpha$-hydroxybutyric acid (150 mM, 15.6 g/L), and the results showed in Figure 8B indicated that the conversion of L-$\alpha$-aminobutyric acid was close to 67%, higher than 46.8% for the wild type.

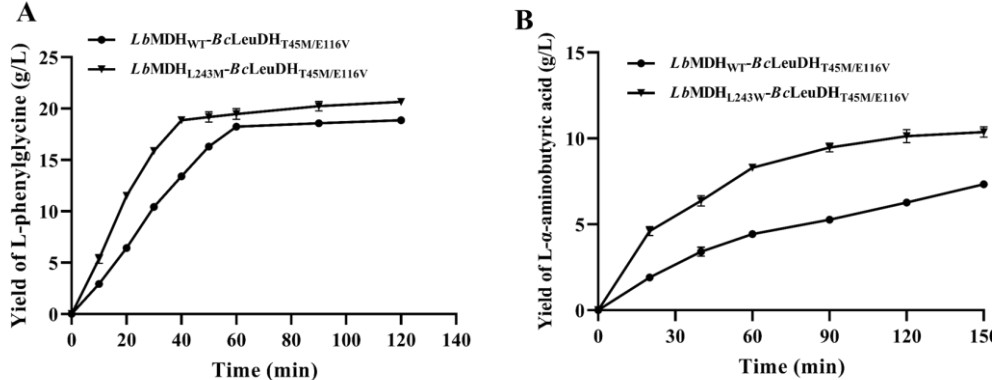

**Figure 8.** Biocatalytic preparation of UAAs. (**A**) the yield of L-phenylglycine obtained using the D-mandelic acid as substrate for the WT and L243M variant; (**B**) the yield of L-α-aminobutyric acid in the hydrogen-borrowing dual-enzyme cascade for the WT and L243W variant. Error bars showed the standard deviation and data were collected from three independent experiments.

## 3. Materials and Methods

### 3.1. Strains, Plasmids, and Materials

The sequence of the D-mandelate dehydrogenases (*Lb*MDH) from *L. brevis* was obtained from GenBank (Gene ID: 4413215) and synthesized by GENEWIZ Biotech Co., Ltd. (Suzhou, China). The genes encoding *Ef*2D2R (Protein ID EGP5414127.1), *Sa*DlacDH (Protein ID OHS90358.1), *Pa*DlacDH (Protein ID PTC35631.1), and *Pa*2D2R (Protein ID EOT12896.1) from *E. faecalis*, *S. aureus,* and *P. aeruginosa* were used as template. The primers used for cloning were listed in Table S1. *E. coli* JM109 was used as the host for gene cloning, and *E. coli* BL21 (DE3) was used as the host for the expression of different enzymes. The plasmid pET-28a (+) purchased from Invitrogen (Carlsbad, CA, USA) was employed as the expression vector. 2 × Phanta Max Master Mix and ClonExpress II One Step Cloning Kit were purchased from Vazyme Biotech Co., Ltd. (Nanjing, China).

The restriction enzymes *EcoR* I and *Hind* III were obtained from TaKaRa (Dalian, China). All of the α-HAs and UAAs were purchased from Sigma–Aldrich (Darmstadt, Germany) or Aladdin Reagent (Shanghai, China).

### 3.2. Site-Directed and Structure Analysis of LbMDH

Site-directed mutagenesis of the *Lb*MDH gene was carried out using the whole-plasmid two-step PCR method [36], and the pET28a-*Lb*MDH plasmid was used as the template. Site-saturation mutagenesis at position L243 was conducted by degenerate codon of NNN (N represents A, T, G or C). The primers used for the mutagenesis were listed in Table S2. All recombinant plasmids were transformed into *E. coli* BL21 (DE3) and sequenced by GENEWIZ Biotech Co., Ltd. (Suzhou, China). Luria–Bertani (LB) broth was used for the growth of *E. coli* and contained kanamycin (50 μg/mL) for plasmid selection.

The structure of *Lb*MDH was built by homology modeling using the free online software SWISS-MODEL with D-mandelate dehydrogenase from *E. faecalis* (PDB entry 5X20) as the template. The docking of substrates and enzymes was done by AutoDock, and the structures of the wild-type *Lb*MDH and its variants were performed by PyMol software. All MD simulations were performed using GROMACS 2018.4 [37] software at 298 K. Amber ff99SB, and GAFF force fields were chosen to describe the proteins and ligands [38], respectively. The MD simulation was performed in several steps, including energy minimization, heating, density equilibration, and 30-ns production MD, with a time step of 2 fs.

### 3.3. Expression and Purification of LbMDH and Its Variants

All the recombinant proteins were expressed by *E. coli* BL21 (DE3) with an N-terminal 6-His tag. Single colonies of these recombinant strains were inoculated into 10 mL LB medium containing 50 µg/mL kanamycin, and cultured at 37 °C with shaking at 180 rpm for 10–12 h as seed liquid. Then, the cells were transformed into 50 mL LB medium with 1% inoculation as well as 50 µg/mL kanamycin and cultured for about 2 h until the OD600 reached 0.6–0.8. Isopropyl B-D-1-thiogalactopyranoside was added to a final concentration of 0.5 mM to induce the protein expression at 16 °C for 10 h. Then, the cell was harvested by centrifugation at 8,000 × q for 10 min.

After washed twice with 0.2 M phosphate buffer (pH 7.0), the cells were suspended again to be sonicated under the ice bath. The cell disruption was centrifuged at 12,000 × g for 20 min at 4 °C to obtain the crude enzyme solution. These solutions were used to purification using $Ni^{2+}$-affinity chromatography on the ÄKTA purifier system by HisTrap column from General Electric Company (Boston, MA, USA). Then, SDS-PAGE analysis and the Bradford protein assay kit were used to check and determine the concentration of the purified enzymes.

### 3.4. Enzyme Activity Assays

As *Lb*MDH uses nicotinamide adenine dinucleotide ($NAD^+$) as the coenzyme, the enzyme activities of *Lb*MDH and its variants were determined by measuring the change of UV absorbance at 340 nm. The unit of enzyme activity was defined as 1 µmol of NAD ($\varepsilon = 6.22$ $mM^{-1}$ $cm^{-1}$) increased per min. The assay mixture contained 6.5 mM D-$\alpha$-hydroxy acid and 1.5 mM NAD in 200 mM $Na_2HPO_4$-NaOH buffer, pH 10.5, and the reaction was started by the addition of limiting amounts of *Lb*MDH. The initial rate of the purified *Lb*MDH and *Bc*LeuDH was determined by measuring the production titer of L-phenylglycine in 30 min.

The optimal temperatures for *Lb*MDH and its variants were determined at pH 10.5 with the temperatures ranging from 25–50 °C. The optimal pH was measured by assaying the enzyme activity using different pH range buffers as follows: citric acid sodium citrate buffer (pH 3.0−6.0, 200 mM), potassium phosphate buffer (pH 6.0−8.0, 200 mM), $NH_3 \cdot H_2O$-$NH_4Cl$ buffer (pH 8.0−10.0, 200 mM), and $Na_2HPO_4$-NaOH buffer (pH 10.0−12.0, 100 mM).

The thermal stability of the enzymes was monitored by enzyme incubation in PBS buffer (50 mM, pH 7.4) at 30 °C and 40 °C for different times. The residual enzyme activities after incubation were measured at 30 °C in $Na_2HPO_4$-NaOH buffer (pH 10.5, 200 mM). The half-lives of thermal inactivation ($t_{1/2}$) of the purified enzymes at 40 °C were calculated using the equation $t_{1/2} = \ln2/k$, and the first-order rate constant, k, was obtained from the slope of a semilogarithmic plot of incubation time versus residual activity. Additionally, to determine the pH stability of *Lb*MDH and its variants, the purified enzymes were incubated in different buffers described above for 12 h at 4 °C. The residual enzyme activity in different pH was measured using the method described for the thermal stability assay.

### 3.5. Determination of Kinetic Characteristics

The Michaelis constant (*Km*) and maximal velocity (*Vmax*) of the wild type *Lb*MDH and its mutations were measured in 200 mM $Na_2HPO_4$-NaOH buffer (pH 10.5) at 30 °C under the condition of one limiting substrate ($\alpha$-HAs or NAD). The *Km* values for $\alpha$-HAs were determined by changing the concentration in the range of 0.12–22 mM, and the *Km* for NAD was measured with the concentration vary from 0.04 to 2 mM. Then, the *Km* and *Vmax* of all substrates were obtained through the Lineweaver Burk method using the program GraphPad Prism 8. The $k_{cat}$ was calculated through the following equation $k_{cat} = Vmax/(E)$, where (*E*) represents the molar concentration of the enzymes.

### 3.6. Circular Dichroism for Structure Analysis and Thermal Melts

Circular dichroism (CD) analysis of *Lb*MDH and L243 mutations was conducted in a J-1700 spectrometer (Jasco, Tokyo, Japan) using a 0.1-cm path-length quartz cuvette. The enzyme samples

were diluted to 0.1 mg/mL in ultrapure water. The far-UV CD spectra were recorded in the range from 190 to 240 nm with a scan speed of 50 nm/min. The melting temperature (Tm) was determined at a rate of 3 °C/min between 20 °C to 70 °C while measuring CD at 222 nm. The Tm of the enzymes were obtained from the first derivative of the thermal melt curves using the program GraphPad Prism 8.

### 3.7. Biocatalytic Preparation of UAAs with a Dual-Enzyme Hydrogen-Borrowing Cascade

The biotransformation system contained 0.9 M $NH_3 \cdot H_2O$-$NH_4Cl$ buffer (pH 9.5), 5 mM $NAD^+$, 0.1 mg/mL *Lb*MDH, 0.3 mg/mL *Bc*LeuDH$_{T45M/E116V}$, 150 mM D-$\alpha$-HAs. Sample aliquots were withdrawn from the reaction mixture to detect the concentrations of substrates and products. The substrates D-$\alpha$-HAs and the intermediate product $\alpha$-keto acids were detected on Agilent LC1260 HPLC system using an Aminex HPX-87H analysis column (Bio-Rad, 300 × 7.8 mm) with 5 mM $H_2SO_4$ as the mobile phase, the UV detector wavelength was set to 254 nm, and the flow rate was 0.5 mL /min at 35 °C. The $\alpha$-amino acids were detected by the HPLC system using the method described by Zhang and Bartolomeo [39,40].

### 4. Conclusions

In summary, we designed a hydrogen-borrowing dual-enzyme cascade for the synthesis of structurally diverse range of aromatic and aliphatic UAAs from D-$\alpha$-HAs. Firstly, we screened a D-mandelate dehydrogenase from *L. brevis* to catalyze D-$\alpha$-HAs to corresponding $\alpha$-keto acids with higher enzyme activities. However, the activity for aliphatic substrates were relatively low. To improve the enzymatic activities towards aliphatic substrates, we rationally engineered the substrate entrance tunnel of *Lb*MDH with the help of sequence and structure analysis. The variant L243W was obtained with higher activity towards D-$\alpha$-hydroxybutyric acid and decreased activity towards D-mandelic acid. Then, site-saturation mutagenesis at L243 was performed. Three important variants with higher enzyme activities and better substrate affinities towards different substrates (L243M, L243W, L243I) were identified. Among the variants, only variant L243M showed improved thermostability compared to the wild type. Structural analysis of these variants showed that mutations of the residues at the entrance of substrate channel of *Lb*MDH significantly affected the size of the channel, thus influencing the enzyme affinities for different substrates. Finally, a hydrogen-borrowing dual-enzyme cascade was designed containing *Lb*MDH and *Bc*LeuDH$_{T45M/E116V}$. With different mutations at L243 of *Lb*MDH, the production of different UAAs greatly improved. However, compared with other reports that use racemic mandelic acid as a substrate to synthesize L-phenylglycine, our conversion towards mandelic acid was not that high. Resch et al. have designed a redox-neutral reaction cascade and achieved 94% conversion for L-phenylglycine from racemic mandelic acid [20], and Fan et al. also report a three-enzyme cascade reaction which can convert 0.2 M rac-mandelic acid to L-phenylglycine with 96.4% conversion rate [22]—whereas, we are the first one to report the biocatalytic preparation of two types of important UAAs including aromatic and aliphatic UAAs from $\alpha$-HAs using a hydrogen-borrowing dual-enzyme cascade.

**Supplementary Materials:** The following are available online at http://www.mdpi.com/2073-4344/10/12/1470/s1, Figure S1: SDS-PAGE analysis of the expression of D-hydroxy acid dehydrogenases from different sources, Figure S2: Optimum pH of D-hydroxy acid dehydrogenase from different sources, Figure S3: Multiple-sequence alignment of D-mandelate dehydrogenase with other D-hydroxy acid dehydrogenase from *Lactobacillus harbinensi* and *E. faecium* by the help of software Clustal X and Esprit 3, Figure S4: Optimum pH of variants of *Lb*MDH, Figure S5: The exponential fitting curves of the data points of thermostability analysis, Figure S6: Structure analysis of *Lb*MDH and the variants by MD simulation, Figure S7: Structure analysis of *Lb*MDH and the variants, Table S1: Primers used for cloning of D-hydroxy acid dehydrogenase, Table S1: Primers used for site-directed and saturated mutagenesis.

**Author Contributions:** Conceptualization, F.L. and J.Z.; methodology, F.L.; software, F.L. and J.Z.; validation, F.L., X.Z., and Z.R.; formal analysis, F.L.; investigation, F.L. and J.Z.; resources, M.X., T.Y., M.S., X.Z., and Z.R.; data curation, F.L. and J.Z.; writing—original draft preparation, F.L.; writing—review and editing, F.L., M.X., T.Y., M.S., X.Z., and Z.R.; visualization, X.Z.; supervision, F.L., M.X., T.Y., and M.S.; project administration, F.L.,

X.Z., and Z.R.; funding acquisition, X.Z. and Z.R. All authors have read and agreed to the published version of the manuscript.

**Funding:** This work was supported by the National Key Research and Development Program of China, (No. 2020YFA0908300), National Natural Science Foundation of China (No. 32071470), Key Research and Development Program of Ningxia Hui Autonomous Region, (No. 2019BCH01002, 2020BFH01001,), Key Research and Development Project of Shandong Province, China (No. 2019JZZY020605), the Project Supported by the Foundation of State Key Laboratory of Biobased Material and Green Papermaking, Qilu University of Technology, Shandong Academy of Sciences (No. KF201907), Program of the Key Laboratory of Industrial Biotechnology, Ministry of Education, China (No. KLIB-KF202009), National First-Class Discipline Program of Light Industry Technology and Engineering, (LITE2018-06), the Project Funded by the Priority Academic Program Development of Jiangsu Higher Education Institutions, Top-Notch Academic Programs Project of Jiangsu Higher Education Institutions.

**Conflicts of Interest:** The authors declare no conflict of interest.

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
