# Peer review of "One-Pot Biocatalytic Preparation of Enantiopure Unusual α-Amino Acids from α-Hydroxy Acids via a Hydrogen-Borrowing Dual-Enzyme Cascade"

_catalysts, doi:10.3390/catal10121470_

Round 1

Reviewer 1 Report

Major revisions

  • The authors claim the idea to combine D-hydroxy acid dehydrogenase and aaDH through hydrogen-borrowing cascade reaction whereas this strategy has already been published in 2010 (Resch, V.; Fabian, W. M. F.; Kroutil, W. Deracemisation of Mandelic Acid to Optically Pure Non-Natural L-Phenylglycine via a Redox-Neutral Biocatalytic Cascade. Adv. Synth. Catal. 2010, 352, 993−997) and used again in 2015 with a mutant enzyme (ref 22 here, but not cited for this purpose). The only novelty here is the mutant able to catalyse the oxidation of non aromatic substrate such as hydroxybutyric acid. So the author must re-write their introduction by placing more emphasis on previous work on this subject and focusing their work/discussion only on the mutant.
  • The reasons why the author chose the W mutations (in addition to the better understood A) are not detailed: this needs to be corrected.
  • The catalytic efficiencies of the mutants are not better than the wild type and the best results are obtained for mandelic acid which is not the main target of the work and for which reference of Resch et al. mentioned above obtained good result. So the authors should shade a little bit more their results in the manuscript including in the abstract.
  • The structural analysis of the mutants have been done with mandelic acid and, according to the authors, aimed to better understand the effect of the mutations to the specific activities. But the authors discussed just briefly the substrate entrance channel, so appropriate analysis would have been molecular dynamics. If the authors did not provide such analysis, they must at least write that this type of analysis would be more accurate to understand the effect of the mutations.
  • Production of phenylglycine: the authors should compare their yields and substrate loading to previous work (at least to results of ref 22 and Resch et al.)

Minor revisions

  • lines 35-36 : reformulate the sentence (no sense at the present state)
  • line 50: typo –Ome? To be reviewed
  • Ref 17 is not appropriate : reviews should be cited here, such 10.1021/acs.chemrev.7b00033
  • Ref 21 : ref 10.1002/cctc.201701366 should be added
  • Table 1: Lactobacillus brevis should be written in italic
  • Table 1: the number of digits must be homogenized between the uncertainties and the main data
  • Kinetics studies: if kinetics studies with NAD+ have been done with mandelic acid as substrate, the kcat should be nearly the same for both of these substrates. It is the case for L243M/L243W and wild type, but not the case for L243I. The author should explain briefly this inconsistency according to the kinetic data obtained (difficulty to saturate the enzyme in second substrate ??)
  • Line 230: (ref) should be modified by the correct reference
  • Line 235/236: proper unit for production efficiency must be used (not min)
  • Gene ID or Uniprot ID of Ef2D2R …Pa2D2R should be provided in M&M

Reviewer 2 Report

an interesting study which can be improved by changing the presentation a bit.

  • format "LbMDH" Lb is not all the time italics; so streamline.
  • L25; italics style of Bacillus cereus to be corrected.
  • L51; cofactors NAD(P)H and O2 ??? NADPH is likely a cosubstrate to drive the enzyme and O2 a substrate while "2" need to be corrected for style.  ... this is the case in similar positions throughout the script eg: "+" in L65, "2" several times in L66 and so on ...
  • Table 1; why was a NADP+ dependent enzyme used - as in the intro it is only NAD-cycle mentioned
  • L135 correct space
  • Fig3; the hight of the 100% value should be either set to same proportion or the 100% compared to original data. Please indicate in legend the 100% = x U/mg for each substrate.
  • Fig5; in B it is not clear from legend what was measured after incubation ... clarify. In Panel C the cut at 100% is not good for presenting data - improve.
  • wording in section 2.5 need to be optimied; structural assay or analysis. what u mean wiht strucutre assay? a reference is needed for the tool used for the docking.
  • Fig 6 improve legend and color code.
  • Fig 7 did u do replicates? - if not add them and show error bars as the Tm differences are so small and thus might be not significant.
  • references need to be corrected for styl and the XXXX to be replaced
  • I miss the supplementary part for judging.

Round 2

Reviewer 1 Report

line 51 : typo "netural" (neutral)

line 55 : typo D-madelate

lines 58-61 : english must be improved ("..., which still presents..." : this is not the right way to express the idea)

lines 62-65 : "high synthesis" is inappropriate, please replace by "high productivity" or "high conversion rates"

ref 24-27 : the authors must cite reviews/books (including pioner work by the Nobel Prize !!)

POint 3/response 3 : Yes, but the catalytic efficiencies are lower despite higher specific activities. The authors must clearly specificy this kcat/Km moderate result, as it is a main point when enzyme improvment is tried.

Lines 216-227 : yes, so these preliminary studies don't allow to explain the effect of the mutation...So again the authors should modify the last sentence to have more moderate words in a sentence explaining that further molecular dynamics targeting the dynamic of the entrance channel should be done to enable hypothesis for the effect of mutation at position 243 on the activities.

lines 378-380: no, the catalytic efficiencies (kcat/Km) are not high. And the products are not pure at all in the reaction mixtures which still contain catalytic amounts of nicotinamide cofactor, buffer salts, enzymes.... The authors should modify this last sentence.

The word biosynthesis (including in the title) is not appropriate. the aujthors have performed biocatalytic reaction. Biosynthesis id more dedicated to synthesis by natural pathways

Reviewer 2 Report

all questions properly corrected; thank you!

Author Response

There are no comments or suggestions for us, thank you very much for your review of the manuscript.